# The Beneficial Effects of Short-Term Exposure to Scuba Diving on Human Mental Health

**DOI:** 10.3390/ijerph17197238

**Published:** 2020-10-03

**Authors:** Arnau Carreño, Mireia Gascon, Cristina Vert, Josep Lloret

**Affiliations:** 1Oceans & Human Health Chair, Institute of Aquatic Ecology, University of Girona, 17003 Girona, Spain; josep.lloret@udg.edu; 2ISGlobal (Global Health Institute Barcelona), Campus Mar, 08003 Barcelona, Spain; mireia.gascon@isglobal.org (M.G.); cristina.vert@isglobal.org (C.V.); 3Department of Experimental and Health Sciences, Universitat Pompeu Fabra (UPF), 08003 Barcelona, Spain; 4CIBER Epidemiología y Salud Pública (CIBERESP), 28029 Madrid, Spain

**Keywords:** scuba diving, mental health, ecosystem health, medication

## Abstract

Exposure to outdoor blue spaces can help improve human health by reducing stress, promoting social relationships, and physical activity. While most studies have focused on the adverse health effects of scuba diving, very few have assessed its health benefits. Moreover, when scuba diving is done in large groups with no diving instructor or pre-dive briefing, negative environmental impacts are generated and negative impacts on human health may also occur due to overcrowding, which may create stress. This is the first study to evaluate the effects of scuba diving on divers’ mental health using their diving practices to estimate the impacts on the ecosystem. In the marine-protected area of Cap de Creus and adjacent areas, we assessed the mental health of 176 divers and 70 beach users (control group) by employing a 29-item version of Profile of Mood State (POMS) questionnaires. According to the parameters associated with reduced environmental impacts, two scuba diving experiences were established. Poisson regression models were performed to assess both the contribution of the activity and diving experiences to POMS scores. Both groups (scuba divers and beach goers) reduced their POMS scores after carrying out the activities. Although no significant differences were found between beach and scuba diving activities, nor between the two different scuba diving experiences, our results showed that subjects with regular medication intake due to a chronic or psychiatric illness had a POMS reduction score significantly higher than other subjects. We conclude that both beach and scuba diving activities have positive effects for human mental health, particularly among subjects with regular medication intake.

## 1. Introduction

A growing number of studies suggest that exposure to natural outdoor environments can help improve human health and wellbeing by reducing stress, encouraging social relationships, and increasing physical activity [1,2,3,4]. The majority of these studies have focused on the benefits of doing outdoor activities in green spaces [5,6], indicating that this improves mental wellbeing and cognitive functioning [7,8]. Several studies also link living close to a green space with better overall mental health, reduced mental distress, lower rates of anxiety, and increased life satisfaction [9,10,11,12].

In recent years, an increasing number of studies have been focusing on the health impacts of outdoor blue spaces (i.e., environments characterized by the presence of water bodies) [13,14,15,16].

Evidence suggests that living or undertaking moderate physical activity near blue spaces such as the coast, lakes, or rivers is associated with better general health and wellbeing. It reduces anxiety and depression [17,18,19], creates opportunities for social interactions [4], and is linked to a better general mood [20,21,22,23]. However, the majority of these studies have focused on the salutogenic effects (i.e., the promotion and maintenance of physical and mental wellbeing) of being close to a water body, rather than being in direct contact with it.

Accordingly, very few studies have assessed the relationship between scuba diving and human health and wellbeing. To date, the majority of studies have focused on the adverse health effects of scuba diving, such as decompression illnesses such as barotraumas and circulation problems, which are related to accidents or malfunctioning of the equipment [24,25,26]. Only a few studies with a very limited number of participants (*N* between 15 and 27) have suggested salutogenic effects, described as a state of wellbeing immediately after diving [27]. For instance, a study carried out on veteran soldiers undertaking diving programs with Deptherapy UK reported an overall improvement in psychosocial wellbeing scores after scuba diving therapy (*N* = 15), including decreased levels of anxiety, insomnia, and depression. Variables were measured using the quantitative 28-Scale General Health Questionnaire and personalized interviews [28]. Deptherapy UK is the first initiative in the world to use scuba diving as therapy and offers specially adapted scuba diving courses for seriously injured veterans of the British Armed Forces, presenting them new mental and physical challenges and empowering them to overcome their illnesses [29]. Another study undertaken through personalized interviews showed positive psychological feelings in disabled people (*N* = 27) through establishing new friendships and developing close, trust-based relationships with their companions [30]. However, the salutogenic effects of water activities such as scuba diving have not yet been explored in the general population.

Despite the potential health benefits for practitioners, scuba diving can generate negative environmental impacts on marine benthic communities under certain circumstances, particularly sessile reef-building organisms such as bryozoans, gorgonians, and corals, among other species [31,32,33]. These impacts are usually worsened when large groups of divers are present without a diving instructor [34,35,36]. Thus, findings from numerous studies have shown that diving with a diving instructor in small groups after receiving a previous ecological briefing significantly reduces the damage divers do to the environment [34,37,38]. Therefore, the positive effects of diving on mental health may be compromised: firstly, by the degradation of natural blue ecosystems [39], as described in other studies assessing the links between the quality of green spaces and their effect on human mental health [40,41,42]; secondly by the presence of large groups of people in the same area. It has been suggested that crowded indoor and outdoor experiences (e.g., in streets, shops, parks, etc.) have no positive impact on mental wellness and may even generate stress because of the proximity to other people, among other factors [43,44,45,46].

We hypothesize that scuba diving may have more benefits for mental health than spending time at the beach. We also hypothesize that diving with an instructor in a small group may be more beneficial, not only for the marine environment, but also the divers’ mental health and wellbeing.

In this context, the present study has two aims: the first is to assess the potential mental health benefits of scuba diving as compared to lying on the beach (control group). The second aim is to compare health benefits between two diving experiences, in order to assess whether diving with a diving instructor in small groups, which is an eco-friendlier practice, can be more beneficial for divers’ mental health than diving without a diving instructor and in large groups.

## 2. Material and Methods

### 2.1. Setting and Study Population

The present study was conducted over two summer seasons (years 2018 and 2019), both on weekdays and weekends in the morning in the marine-protected area of Cap de Creus and the adjacent areas (municipalities of Roses and Llançà, Catalonia, Spain) (Figure 1). The marine protected area of Cap de Creus hosts and protects a multitude of Mediterranean emblematic and endangered species such as the dusky grouper (*Epinephelus marginatus*), the brown meagre (*Sciaena umbra*), red coral (*Corallium rubrum*), and gorgonians (*Paramuricea clavata*), which are highly valued by scuba divers. This area is visited by tourists from around the world and has many marine and land activities operated by various diving centers (among other stakeholders), which do more than 60,000 dives per year [33].

Considering that the first aim of the study was to evaluate the mental health benefits of scuba diving, we used beach users (i.e., participants conducting a leisure activity, which involved interaction with a blue space, but not scuba diving) as the control group. Thus, the study sample included scuba divers (the main interest group), who were divided into two groups (A and B) according to two sets of scuba diving experience characteristics, and beach users (control group). We approached potential participants about to start their activity (i.e., lying on the beach or scuba diving) either on the beach itself, or by boarding a scuba diving boat. We explained the study to them and asked if they were willing to participate. If they accepted, we explained that the objective of the study was to find out how spending some time on the beach/going scuba diving can affect human mental health. Their written consent was obtained, and they were given the first questionnaire in that moment. We made sure that all participants understood every question correctly. We distributed the first questionnaire then returned 1 h later, after their exposure to the blue space, to give participants the final questionnaire.

In order to take part in the study, beach users had to be intending to stay on the beach for at least 1 h. The inclusion criteria required participants to be over 12 years old. Participants who went scuba diving were told to carry out their diving session (approximately 1 h) as per usual. The study was approved by the ethics and biosecurity committee of the University of Girona (CEBRUdG).

### 2.2. Scuba Diving Experiences

In order to compare different scuba diving experiences, we boarded scuba diving boats from four different diving centers operating in the Cap de Creus marine protected area. We established two groups according to three characteristics of the diving experience: (i) the size of the group (15 or less divers vs. more than 15 divers in the diving boat); (ii) diving (or not) with an instructor; (iii) receiving (or not) a complete pre-dive briefing in which the particularities of the site, tips about safety, and the importance of respecting the environment and marine species were explained. Based on these criteria, we established two different groups. On one hand, group A, which included divers who were more respectful towards the environment, as they complied with at least 2 of the 3 previously mentioned criteria, criterion (i) (diving in a group of 15 divers, or less) being mandatory. On the other hand, group B included participants who respected the environment less (i.e., participants dived in larger groups of divers (>15), did not dive with an instructor, or did not receive a complete pre-dive briefing) (Table 1).

### 2.3. Individual Characteristics and Psychological Evaluation

All participants in the study were asked to answer a questionnaire immediately before and after participating in their respective activities (approximately 1 h of difference between both questionnaires). Questionnaires distributed before the activity included the following information: (i) Individual characteristics (e.g., gender (i.e., female, male); age (i.e., years of age); children under 18 (yes, no); living as a couple (yes, no); current residence with sea views (yes, no); educational level (incomplete primary education, completed primary education, completed secondary education, completed higher education); purchasing power (significant financial difficulties, with certain financial difficulties, neither living comfortably nor having financial difficulties, living comfortably, does not know/does not want to answer); resident of the municipality (defined as local, summer season resident, or tourist); visiting because it was a marine-protected area (yes, no); own perception of their health (1–4), and regular medication intake (yes, no)). Medication intake was defined as: participants who took medication to treat chronic or psychiatric illnesses. Medication use included: anti-depressants, anti-inflammatories, immunosuppressants, pain killers, and kidney protectors). (ii) How they felt the previous day and how they had slept (e.g., easy to sleep the previous night (1–5); disturbed sleep the previous night (1–5); woke up earlier than usual (1–5); overall quality of sleep (1–5); happiness the previous day (1–10); and anxiety the previous day (1–10)). Questionnaires distributed after the activity included the following information: (i) Activity (scuba diving or spending time on the beach) characteristics (e.g., uncomfortable because of people (1–5); uncomfortable because of pollution (1–5); uncomfortable because of fishing gear (1–5); happiness while doing the activity (1–5); active while doing the activity (1–5); safe while doing the activity (1–5), or feeling that they were in a marine-protected area (1–5)) (Table 1).

To assess the contribution of the activities to mental health we used the Profile of Mood State (POMS) in both questionnaires (before and after the activity). Originally developed by McNair, Loor, and Droppleman in 1971 [47], this is a well-established, academic, and clinically validated measure of psychological distress and mental wellbeing consisting of 65 mood state questions grouped into 6 categories. POMS questionnaires have been used in a similar study that assessed the effects of medium-term exposure to blue and urban spaces on the mental health of participants undertaking various days of walking routines [23]. For this study, we used the 29-item version established by Fuentes et al. (1995) [48], which divides mood states into 5 mood categories and explains 92.9% of the covariance of the original questionnaire: tension/anxiety (TA), depression (D), anger/hostility (AH), fatigue (F), and vigor (V). Responses for each item were rated on a five-point scale ranging from “Not at all” to “Very much”. The total score for POMS before and after the activity was calculated using the following formula: [(TA) + (D) + (AH) + (F) – (V)]. Following this formula, higher POMS scores indicated worse psychological distress. In order to avoid negative numbers when fitting the model, we added 100 to the POMS final score (before and after the activity) [48]. In addition, we also collected information among divers regarding their diving experience (e.g., if they had a diving certificate, years of experience, etc.). Weather and air and water temperatures were also recorded.

A total of 76 beach users agreed to participate in the study. Of these, 70 answered the POMS section of the questionnaire correctly and were included in the study. We recruited 181 participants in the scuba diving group. Of these, 176 (73 from diving experience A and 103 from diving experience B) answered the POMS section of the questionnaire correctly and were included in the study.

### 2.4. Data Analysis

Participants who failed to answer the POMS section of the questionnaire correctly were discarded from the study, as POMS results were the main outcome of the study.

As the age variable had a significant number of missing values (44% in the control group and 36% in the diving group), imputation models for each group were run in order to predict age. The variables used to predict the age of the participants were those related to their sociodemographic characteristics and economic status (age, gender, educational level, income, residence, children below 18 years old, living as a couple, and their own perception of health). The age predicted for each subject was then substituted if the age variable was missing, whilst prediction was not used if age was given. We were unable to predict age for 10 participants because some of the predictive variables were also missing (Table 1). Other variables were not predicted as the number of missing values was very low (<5%) and did not influence the models run.

T-test comparison between groups was used to determine significant differences between descriptive continuous variables (i.e., age) and binary variables (i.e., gender); polychorical tests were performed to determine significant differences in categorical variables with more than two categories (i.e., purchasing power).

Poisson regression models were used to assess the association between the activity of interest (i.e., scuba diving) and mental health. Based on previous literature, the following variables of adjustment were included in the models: gender, age, having children under 18, current residence with sea views, regular medication intake, previous night’s sleep quality, feeling safe while doing the activity, feeling uncomfortable because of pollution, and the POMS score before doing the activity. Data were analyzed using STATA MP Software (version 15; StataCorp LLC, 4905 Lakeway Drive, College Station, TX, USA).

## 3. Results

### 3.1. Differences between Spending Time on the Beach and Scuba Diving

Characteristics of the study population are shown in Table 2. In total, 76 beach users from different European countries, ranging in age from 21 to 80 years (*N* = 39 female and *N* = 30 male), agreed to participate in the study. Of these, 70 answered the POMS section of the questionnaire correctly. We recruited 181 participants in the scuba diving group, ranging from between 13 and 71 years old (*N* = 51 female and *N* = 122 male) from different European countries. One hundred and seventy-six answered the POMS section of the questionnaire correctly and were included in the models.

The T-test showed significant differences between study groups (scuba diving and beach attendees) in the following variables: gender (43% of men on the beach vs. 71% of men scuba diving); age (mean age 50 years old in the beach group and 43 years old in the scuba diving group); regular medication intake (10% vs. 3% of the participants, respectively); uncomfortable because people and uncomfortable because of pollution (the prevalence of both significantly higher in the beach group); anxiousness the day before (score 3.03 in the beach group and 2.27 in the scuba diving group); active while doing the activity (score 3.80 vs. 4.15, respectively), came to the area because they knew it was a marine-protected area (11% vs. 51%, respectively), and had the feeling they were in a marine-protected area (1.9 vs. 3.5) with significantly higher scores in the scuba diving group (Table 2).

The T-test also showed statistically significant (*p* ≤ 0.05) differences in the POMS scores categories of anger and depression (significantly lower in the scuba diving group both before and after the activity,); fatigue after the activity (3.17 for the beach group and 2.03 for scuba diving group); and in POMS total score after the activity (97.14 and 93.18, respectively) (Table 2).

### 3.2. Differences between Scuba Diving Experiences

The characteristics of scuba divers according to their diving experience in groups A and B (Table 1) are summarized in Table 3. Specific data related to diving experience, depth, and sea and air temperatures are also shown. The T-test shows significant differences between diving experiences in the variables of age, the mean age being 39 years in group A and 44 years in group B; sea water temperature, 23 °C in group A and 24 °C in group B; and in the POMS Tension score before the activity, which was significantly higher for group A (5.07 vs. 3.75 in group B).

### 3.3. Contribution of both Activities to Mental Health

We observed that three variables (POMS before the activity for total scores and for each category (tension–anxiety, fatigue, anger, depression, and vigor), regular medication intake and children under 18 (data not shown)) were statistically significantly associated with the total POMS score after the activities, as well as with the final score of specific POMS categories. However, no significant contribution of scuba diving to the POMS final score was observed, neither from diving experience A (more respectful towards the environment) nor diving experience B (with more associated environmental impacts), compared to the control group, beach attendees (Table 4). Although there was a tendency to a reduction in the POMS total score after scuba diving (both experience A and B compared to beach activity Incidence Rate Ratio (IRR) = 0.97, 95%CI (0.94–1.01); IRR = 0.98, 95%CI (0.95–1.02), respectively), this tendency was not statistically significant.

As indicated above, we observed that the regular medication intake variable was a strong predictor of the total POMS score after the activity [IRR = 0.91 (95%CI = 0.85, 0.97); *p* < 0.01] mainly for its effect on the POMS vigor category [IRR = 1.26 (95%CI = 1.08, 1.48); *p* < 0.01] (Table 5). In total, there were 13 subjects who took regular medication, seven in the beach activity and six in scuba diving. Figure 2 shows total POMS scores before and after the activity according to regular medication intake; significant differences were observed in total POMS score before the activity, which was higher among subjects with regular medication intake. In this group, we observed a stronger reduction in the total POMS score after doing the activity, regardless of which activity (i.e., scuba diving or spending time on beach) compared to those who did not take regular medication (Figure 2). In fact, evidence of an interaction between regular medication intake and the activity performed was not observed (*p* > 0.05).

## 4. Discussion

Interestingly, the innovative finding of our study was not among our initial scientific objectives. This was the observation that after conducting an activity in a blue space (i.e., scuba diving or spending time on the beach), a sharper decrease in the total POMS score could be observed among participants taking regular medication compared to those who did not take regular medication, regardless of whether the activity was scuba diving or spending time on the beach. In addition, we observed that participants showed reduced psychological distress after just one hour of activity, regardless of whether the activity was scuba diving or spending time on the beach, thus providing further evidence to support the concept that exposure to blue spaces contributes to improving human wellbeing [14,16,49]. Moreover, results showed that both scuba diving and spending time on the beach may share the same potential for benefitting human mental health after relatively short exposure; results that align with previous studies assessing the effects of short-time exposure to blue spaces [14,50]. However, we did not observe greater mental health benefits among scuba divers compared to beach users, which was our initial hypothesis. Furthermore, no significant differences were observed among scuba divers based on their diving experiences: group A was more respectful towards the environment; while group B was associated with causing environmental impacts. Although previous studies in green spaces indicate that more preserved areas (i.e., presence of trees, large extensions of fields, etc.) are linked to better mental health amongst users [41,42], we did not observe such results in a preserved blue space (MPA of Cap de Creus). This is perhaps due to participants only being exposed to a blue space for a very short time (1 h), not enough time for the contribution of a preserved landscape or ecosystem to influence participants’ mental health. In this regard, further studies are required to assess medium and long-term exposure to preserved blue spaces and evaluate their effect on mental health. Although it was not among our initial scientific objectives, and despite the limitation of having only 13 subjects with regular medication intake (*N* = 7 in the beach group and *N* = 6 in scuba diving group), we observed that regular medication intake was a strong predictor of the outcome (POMS after the activity score). At baseline, subjects with regular medication intake had worse mental health scores before the activity. This was expected, as regular medication intake is an indirect measure of health status and, therefore, a potential estimator of mental wellbeing. Indeed, people with chronic illnesses tend to score worse in mental health outcomes [51,52,53]. After doing the activity, the mental health of these subjects improved, as did the mental health of participants not taking regular medication, both groups reaching similar scores. Therefore, although all participants (scuba divers and beach users) experienced significant health benefits after the activity, the health benefits were greater among people with regular medication intake. We assume this is a sector of the population that has poorer mental health.

Regarding POMS categories, these pre- and post-activity differences were only observed for the vigor category, which reflects mood states such as “full of energy” or “brave”. Therefore, the effect of the activity was stronger in participants with regular medication intake. They felt more stimulated after performing the activity in a blue space, which possibly gave them a mental status that made their illness go temporarily unnoticed, and a willingness to enjoy the space and the activity they were doing. In previous studies, similar benefits for health and wellbeing were reported in patients recovering in hospital from surgery. The patients who had views of a green space recovered faster, received fewer potent analgesics, and complained less than those who only had views of a brick wall [54].

To our knowledge, this is the first experimental study reporting the health benefits of short-term exposure to blue spaces among subjects with regular medication intake (i.e., indicator of chronic illnesses). Previously, Morgan et al. (2018) examined the mental health benefits of long-term exposure in a study by on war veterans participating in a scuba diving course (*N* = 10). This study, however, had no control group and used a different questionnaire (the General Health Questionnaire-28, completed during and after the diving course) to assess mental health [28]. The results pertain to medium/long-term exposure to blue spaces (on completion of a scuba diving course) among disabled war veterans. Our findings, on the other hand, suggest similar effects in the short-term among the general population, particularly individuals with regular medication intake. Therefore, the present study and that of Morgan et al. (2018) may be forerunners in providing the first opportunity to investigate the contribution blue spaces make to the mental health of patients with chronic physical or psychiatric illnesses. Along these lines, further studies need to be undertaken on large groups of patients which assess both the potential short- and long-term benefits of doing maritime recreational activities, especially scuba diving, on mental health and wellbeing. Finally, if the positive effects of scuba diving on practitioners’ mental health are confirmed, initiatives similar to Deptherapy U.K. could be set up in the Mediterranean where the warmer sea allows for longer periods of scuba diving throughout the year.

### 4.1. Blue Spaces, Wellbeing, and Ecosystem Health

Regarding the mental health effects of scuba diving, differences in diving experiences between Group A and B were not observed. However, it is well-known that diving in small groups, with a diving instructor and receiving a complete briefing prior to diving, significantly reduces the damage divers cause to marine benthic species [34,37,38]. Therefore, conducting sustainable scuba diving practices that respect the environment is a strategy that needs to be promoted, regardless of whether health benefits exist or not. In the context of planetary health [55,56,57], this strategy will guarantee both preserving the health of ecosystems and promoting scuba divers’ health. The quality of blue spaces has been broadly linked to increased usage because of the attraction blue spaces hold for recreationists, with more activities being developed in these spaces and an increase in the population’s wellbeing, especially via the mechanism of increased physical activity [4,13,58]. However, more users usually mean more environmental damage [33]; therefore, there is a need to balance preserving a healthy environment with promoting a healthy population.

### 4.2. Strengths and Limitations of the Study

Despite the novelty of this research, its strengths and limitations must be acknowledged. Firstly, we found no mental health benefits related to scuba diving when compared to the control group (beach users), nor differences between scuba diving experiences. A possible explanation for these results could be that both groups (scuba diving and beach users) show similar characteristics when exposed to blue spaces. Beach goers were chosen as the control group because they were subjects readily available for interview by the study researchers and had potentially similar characteristics to those scuba diving (i.e., people on holiday, doing a relaxing activity in a blue space). As a result, the control group received the same input and the same benefits from being close to a blue space. Nevertheless, an additional control group in a closed or urban space with participants with similar socio-economic characteristics should have been created to determine the magnitude of mental health benefits, both in scuba diving and beach activities. In addition, it would also be interesting to compare these groups with others doing activities in green spaces (i.e., forest walking guided tours, recreational walks, interactive study of local fauna and flora, etc.), as many studies have also reported beneficial outcomes of green spaces for mental health [40,59,60]. Moreover, we assumed that every subject was its own control before participating in the activity and failed to take into account the number of days each subject had been exposed to the blue space previous to the interview. Therefore, further studies need to be carried out on a number of subjects with known previous exposure to blue spaces, and a control group that has had no access to a blue space. However, we expect that finding participants with similar socio-economic characteristics, and who live in a coastal town, but do not have access to a blue space will be difficult.

Secondly, this study was not designed to evaluate whether people suffering from certain medical conditions (using medication intake as proxy) had different health outcomes as a result of the activity. Additional information regarding medication intake was requested from scuba divers a posteriori; for this reason the study contained only 13 participants with regular drug intake.

Thirdly, this study had the logistic difficulty of having to board a scuba diving boat and ask scuba divers to participate, answer questions, and assure that questionnaires were filled in just before and immediately after the dive, taking into account that divers have to prepare their equipment and emerge wet from the water. Despite this difficulty, the divers’ response was good and their willingness to participate in this study surprisingly positive, with over 90% of those invited to participate in the study accepting.

Fourthly, one of the (unexpected) limitations we encountered was the lack of response regarding participants’ age both among scuba divers and beach attendees; therefore, age had to be imputed. An external review of the questionnaires showed no difficulties in identifying or answering the age field, so we could not determine the reasons for the subjects’ unwillingness to provide their age. Regarding the validity of our prediction, we observed statistically significant differences between scuba diving and beach groups before and after age prediction, which provides confidence in the prediction procedure followed. The fact that the average age was lower among scuba divers is not surprising, as scuba diving is a low to moderate impact physical activity that requires a minimum level of fitness. Regarding gender, this fails to explain differences between activity groups. It appears that scuba diving is undertaken by men more than women. In this regard, a strategical approach should be taken to reduce the gender gap and encourage more women to participate in this type of activity.

Finally, in the present study, we only assessed the short-term benefits of short-term exposure in a blue space. However, it would be interesting to further explore how long these health benefits continue after short-term exposure, or how constant this short-term exposure needs to be in order to give sustained benefits over time.

## 5. Conclusions

Both beach and scuba diving activities have positive effects for human mental health, particularly among subjects with regular medication intake, despite no significant differences being found between the activities (both contribute equally to benefitting the mental health of participants).

Although no significant differences regarding wellbeing have been found between diving experiences, diving in small groups, with a diving instructor, and after receiving a complete briefing is expected to have significantly less environmental impact. Therefore, a strategy that balances both the health of the scuba divers and the health of the environment must be promoted in order to enhance diver’s health and preserve the marine environment.

Further studies are required to assess the potential benefits of activities related to blue spaces (e.g., scuba diving, beach walking, etc.) on mental health and wellbeing in the short, medium, and long term. Moreover, the results of this research point to the need for more studies focusing on the potential mental health benefits of exposure to blue spaces (i.e., spending time on the beach or scuba diving) among patients suffering from illnesses, particularly chronic illnesses. These could be compared to exposure to green or urban spaces and/or another group of healthy participants. This study adds to the cumulative evidence attesting the health effects of blue spaces; although further investigation is needed to overcome the existing limitations, our results support the idea that health programs should use blue spaces as therapeutic environments.

## Figures and Tables

**Figure 1 ijerph-17-07238-f001:**
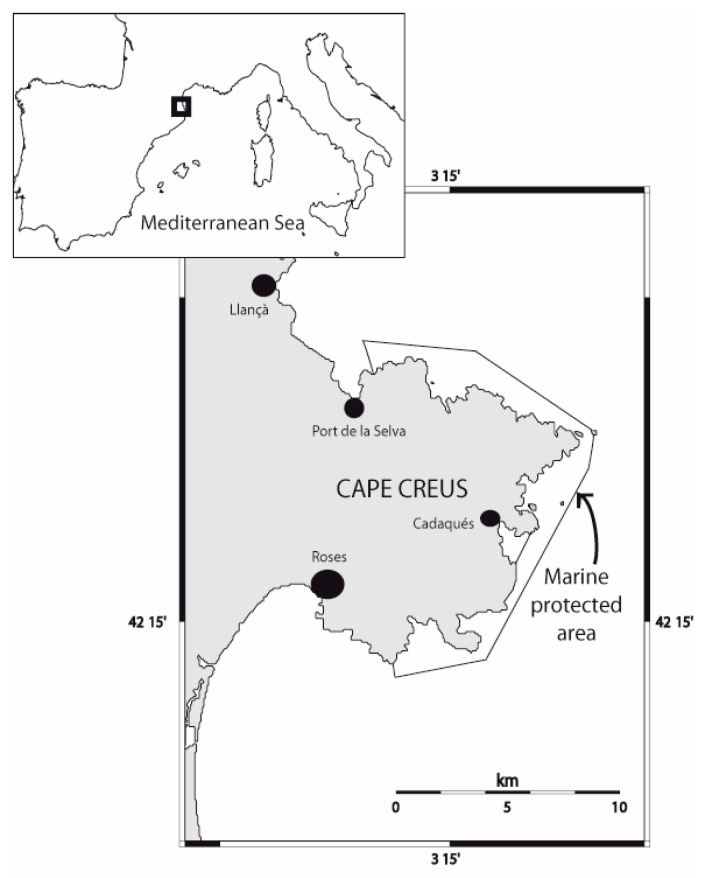
The marine protected area of Cap de Creus.

**Figure 2 ijerph-17-07238-f002:**
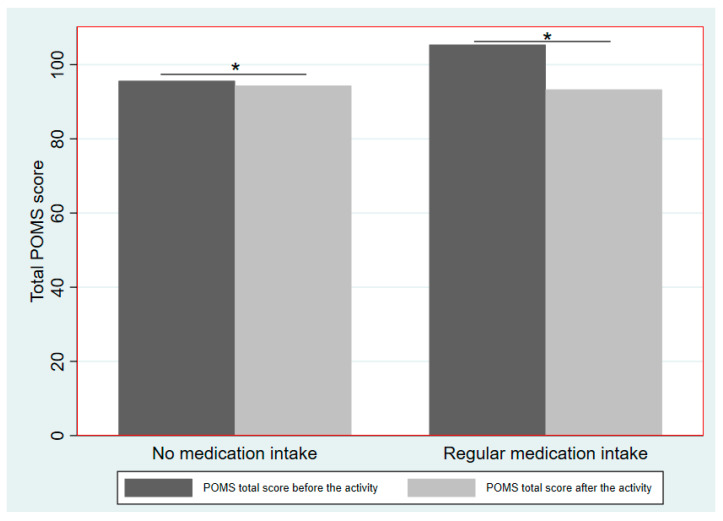
Differences between Profile of Mood States (POMS) total scores according to regular medication intake (yes, no), grouped by medication. * *p* < 0.05.

**Table 1 ijerph-17-07238-t001:** Comparison of diving experiences. Diving experience A complied with at least 2 of the 3 criteria.

Diving Experience A	Diving Experience B
Groups of 15 divers or less (mandatory criterion)	Groups of more than 15 divers
Diving with a diving instructor	Diving without a diving instructor
A complete (>10-min) pre-dive briefing given	A short (<10-min) pre-dive briefing given

**Table 2 ijerph-17-07238-t002:** Descriptive data of the study population ^a^.

Variable/Activity	Beach (*N* = 70)			Scuba diving (*N* = 176)			
	*N*	Mean (SD)	Min–Max	*N*	Mean (SD)	Min–Max	Statistically Significant *p*-Value ≤ 0.05
**Age (years)**	69	50.4 (15.2)	21–80	167	41.5 (13.3)	13–71	*
**Gender (%,** **male)**	69	43	0–1	173	71	0–1	*
**Resident of the municipality (%)**	70			175			
Tourist	40	57		148	84.6		
Seasonal	18	26		19	10.9		
Local	12	17		8	4.57		
**Current residence with sea views (%)**	70	53	0–1	176	60	0–1	
**Children under 18 (%)**	70	24	0–1	176	32	0–1	
**Living as a couple (%)**	70	83	0–1	172	71	0–1	
**Studies (%)**	69			176			
Incomplete primary education	0			2	1.1		
Completed primary education	8	11.6		9	5.1		
Completed secondary education	29	42.0		64	36.4		
Completed higher education	32	46.4		101	57.4		
**Purchasing power (%)**	69			174			
A lot of financial difficulties	1	1.4		0	0.0		
With certain difficulties	5	7.2		8	4.6		
Neither living comfortably nor having difficulties	16	23.2		33	19.0		
Living comfortably	44	63.8		128	73.6		
Does not know/does not want to answer	3	4.3		5	2.9		
Usual residence close to a blue space (15min walking) (%)	70	67	0–1	176	57	0–1	
Came because of it being a marine-protected area (%)	66	11	0–1	176	51	0–1	*
Regular medication intake (%)	69	10	0–1	176	3	0–1	*
Easy to sleep the previous night	69	3.9 (1.1)	1–5	176	3.8 (1.1)	1–5	
Disturbed sleep the previous night	69	2.6 (1.3)	1–5	176	2.3 (1.2)	1–5	
Woke up earlier than usual	70	2.8 (1.5)	1–5	176	2.5 (1.5)	1–5	
Overall quality of sleep the previous night	70	3.5 (1.4)	1–5	176	3.6 (1.1)	1–5	
Own perception of their health	70	3.0 (0.6)	1–4	176	3.2 (0.6)	1–4	
Happiness yesterday	70	8.0 (1.9)	1–10	176	8.0 (1.5)	1–10	
Anxiousness yesterday	70	3.0 (2.5)	1–10	176	2.3 (2.0)	1–10	*
People in their group	69	2.7 (2.5)	1–15	162	3.1 (1.0)	1–6	
Uncomfortable because of people	69	1.8 (1.0)	1–5	176	1.3 (0.7)	1–5	*
Uncomfortable because of pollution	69	1.8 (1.1)	1–5	176	1.5 (0.9)	1–5	*
Uncomfortable because of fishing gears	69	1.1 (0.4)	1–3	176	1.2 (0.7)	1–5	
Happiness while doing the activity	69	4.3 (0.9)	0–5	176	4.4 (0.8)	0–5	
Active while practicing the activity	69	3.8 (1.0)	0–5	176	4.1 (1.1)	0–5	*
Safe while practicing the activity	69	4.5 (0.8)	1–5	173	4.7 (0.6)	1–5	
Feeling that they are in a marine-protected area	69	1.9 (1.1)	0–5	173	3.5 (1.3)	0–5	*
**POMS results**							
**Anger–hostility score**							
Before activity	70	2.4 (4.6)	0–21	176	1.1 (2.2)	0–15	*
After activity	70	1.8 (3.9)	0–19	176	0.6 (1.5)	0–12	*
**Tension–anxiety score**							
Before activity	70	4.6 (4.9)	0–21	176	4.3 (3.0)	0–13	
After activity	70	3.4 (4.4)	0–17	176	3.0 (2.6)	0–12	
**Fatigue score**							
Before activity	70	3.0 (4.0)	0–17	176	2.4 (2.7)	0–13	
After activity	70	3.2 (3.7)	0–15	176	2.0 (2.5)	0–11	*
**Depression score**							
Before activity	70	2.2 (4.7)	0–20	176	0.9 (2.1)	0–17	*
After activity	70	1.7 (4.2)	0–20	176	0.4 (1.3)	0–12	*
**Vigor score**							
Before activity	70	13.6 (5.0)	2–24	176	13.5 (5.1)	0–24	
After activity	70	13.0 (4.6)	2–24	176	12.8 (5.1)	0–24	
**Total POMS score**							
Before activity	70	98.6 (17.6)	76–168	176	95.1 (10.6)	76–140	
After activity	70	97.1 (14.6)	76–160	176	93.2 (8.8)	76–134	*

^a^ T-test comparison between groups was used to determine significant differences between descriptive continuous variables (i.e., age) and also binary variables (i.e., gender) and Profile of Mood States (POMS) scores; polychorical tests were performed to determine significant differences in categorical variables with more than two categories (i.e., purchasing power). * *p* < 0.05.

**Table 3 ijerph-17-07238-t003:** Description of the scuba diving groups according to diving experiences ^a,b^.

Variable	Diving Experience A (*N* = 73)			Diving Experience B (*N* = 103)			
	*N*	Mean (SD)	Min–Max	*N*	Mean (SD)	Min–Max	Statistically Significant *p*-Value ≤ 0.05
**Age (years)**	69	38.5 (14.9)	13–71	98	43.9 (12.0)	16–69	*
**Gender (%, male)**	72	60	0–1	101	70	0–1	
**Experience (number of years) (%)**	73			103			
1 to 5	33	45.2		27	26.2		
6 to 10	10	13.7		18	17.4		
11 to 20	13	17.8		31	30.1		
more than 20	17	23.3		27	26.2		
**Dives per year (%)**							
First time	2	2.7		0	0.0		
1 to 10	28	38.4		34	33.0		
11 to 20	17	23.3		29	28.2		
more than 20	26	35.6		40	38.8		
**Depth**	73	2.3 (1.1)	0–4		2.3 (1.1)	0–4	
**Sea temperature (°C)**	73	23.0 (2.9)	18–26	103	23.8 (2.2)	19–26	*
**Air temperature (°C)**	73	26.2 (2.7)	18–29	103	26.4 (2.7)	22–31	
**Came because it was a marine-protected area**	73	0.6 (0.5)	0–1	103	0.4 (0.5)	0–1	
**Felt they had dived in a marine-protected area**	72	3.6 (1.3)	1–5	101	3.5 (1.2)	1–5	
**POMS results**							
**Anger–Hostility score**							
Before activity	73	0.8 (1.4)	0–6	103	1.4 (2.6)	0–15	
After activity	73	0.5 (0.9)	0–4	103	0.6 (1.8)	0–12	
**Tension–Anxiety score**							
Before activity	73	5.1 (2.9)	0–13	103	3.7 (3.0)	0–13	*
After activity	73	3.4 (2.8)	0–12	103	2.7 (2.5)	0–12	
**Fatigue score**							
Before activity	73	2.4 (2.7)	0–12	103	2.3 (2.8)	0–13	
After activity	73	2.2 (2.5)	0–11	103	1.9 (2.6)	0–11	
**Depression score**							
Before activity	73	0.7 (1.5)	0–6	103	1.0 (2.4)	0–17	
After activity	73	0.3 (0.8)	0–4	103	0.5 (1.6)	0–12	
**Vigor score**							
Before activity	73	14.0 (5.3)	0–23	103	13.2 (4.9)	0–24	
After activity	73	13.4 (5.2)	0–24	103	12.4 (5.1)	0–24	
**Total POMS score**							
Before activity	73	94.9 (9.2)	78–119	103	95.3 (11.5)	76–140	
After activity	73	93.0 (8.1)	76–113	103	93.3 (9.4)	76–134	

^a^ Diving experience A = complied with 2 out of 3 following criteria: diving group < 15 pax (mandatory criterion), dived with diving instructor and had an ecological briefing; diving experience B = complied with 2 out of 3 following criteria: diving group > 15 pax, dived without diving instructor, did not receive an ecological briefing; ^b^ T-test comparison between groups was used to determine significant differences between descriptive continuous variables (i.e., age) and also binary variables (i.e., gender); polychorical tests were performed to determine significant differences in categorical variables with more than two categories (i.e., experience). * *p* < 0.05.

**Table 4 ijerph-17-07238-t004:** Poisson regression model for total POMS and the different POMS categories (tension–anxiety, fatigue, anger, depression, and vigor) scores after activities, Incidence Rate Ratios (IRR) and *p*-values (*p*) ^a^. Beach as activity of reference ^b^.

Outcomes and Variables	IRR (95%CI)	Statistically Significant *p*-Value ≤ 0.05
**Total POMS score**		
Diving experience A	0.97 (0.94–1.01)	0.18
Diving experience B	0.98 (0.95–1.02)	0.26
**Tension–anxiety score**		
Diving experience A	1.21 (0.91–1.60)	0.19
Diving experience B	1.16 (0.89–1.52)	0.27
**Fatigue score**		
Diving experience A	0.79 (0.53–1.18)	0.25
Diving experience B	0.76 (0.54–1.08)	0.13
**Anger–hostility score**		
Diving experience A	1.12 (0.53–2.36)	0.78
Diving experience B	0.90 (0.47–1.74)	0.76
**Depression score**		
Diving experience A	0.67 (0.25–1.83)	0.44
Diving experience B	0.75 (0.32–1.74)	0.50
**Vigor score**		
Diving experience A	1.05 (0.94–1.16)	0.40
Diving experience B	1.01 (0.91–1.11)	0.91

^a^: Poisson regression model adjusted by gender, age, children under 18, current residence with sea views, regular medication intake, previous night sleep quality, safe while doing the activity, and uncomfortable because of pollution; ^b^: T-tests for paired data were used to determine differences between POMS final scores in variables established as significant by Poisson regression models.

**Table 5 ijerph-17-07238-t005:** Association between medication intake for total POMS and the different POMS categories scores ^a^.

Regular Medication Intake Contribution to:	IRR (95%CI)	Statistically Significant *p*-Value ≤ 0.05
Total POMS score	0.91 (0.85, 0.97)	<0.01 *
POMS categories		
Tension	0.78 (0.49, 1.23)	0.28
Fatigue	0.73 (0.38, 1.41)	0.35
Anger	0.50 (0.15, 1.70)	0.269
Depression	0.86 (0.22, 3.28)	0.82
Vigor	1.26 (1.08, 1.48)	<0.01 *

^a^ Poisson regression model adjusted by gender, age, children under 18, current residence with sea views, regular medication intake, previous night sleep quality, safe while doing the activity, and uncomfortable because of pollution. * *p* < 0.05.

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
