# Peer review of "The Beneficial Effects of Short-Term Exposure to Scuba Diving on Human Mental Health"

_ijerph, 2020, doi:10.3390/ijerph17197238_

Round 1

Reviewer 1 Report

Thank you for the opportunity to review your interesting paper, which appears to provide an interesting new perspective on the wellness benefits of blue spaces. Despite the non-significant outcomes that did not thus support your initial expectations, the findings nonetheless add to the growing body of research addressing the various psychosocial and physiological benefits to be gained through immersion in natural environments, whether they be green, blue, brown (e.g. deserts) or white (e.g. snow). I found only a few areas where corrections are required as follows: ABSTRACT Semi-colon used incorrectly throughout; replace all instances with a comma PAPER L 52, delete d from decreased; should be decrease L 189, Table 2. The headings for Mean (SD) are relevant only for the first line (age), with all following rows reporting percentages. I suggest modifying with this in mind. E.g., could insert another heading row following the age row to report the detail more correctly. I also suggest splitting demographic data reporting percentages away from the data reported as measures (e.g. POMS), where the p values column becomes relevant. Ll 310-312 In addition to (or instead of) comparison with people in urban areas, it could be useful to compare with guided/non-guided activities in green spaces, which are also known to provide beneficial wellness outcomes.

Reviewer 2 Report

It is my pleasure to review this interesting manuscript. The major purpose of this manuscript was to examine the examine the effects of scuba diving on divers’ mental health. However, I have several concerns about your study. Please consider the following comments based upon my review of this manuscript. I hope that you will find them helpful.

Abstract:

  1. In your conclusion subsection, it would be better if you can highlight the major research findings and research contribution/practical implications in the current study.

Introduction:

  1. Does the review of literature provide a comprehensive and concise rationale for the study purpose? The introduction describing previous research evidence is not inclusive. A stronger introduction should be given.
  2. Please list your research hypotheses in this study.

Methods:

  1. More information is needed when you introduced the individual characteristics and psychological evaluation such as POMS.
  2. How about the validity and reliability of POMS assessment in your current study?
  3. Is it possible to expand and explain your descriptive results?
  4. Can you report/evaluate your effect sizes based on your study findings?
  5. How did you deal with the missing data in your study if it is a case?

Discussion and Conclusions:

  1. Can you relate your research findings to previous research evidence in order to highlight the major contribution of your study? We cannot identify whether your research findings strongly link to current literature or theory/practice based on your current version.
  2. There are several formatting violations in the reference list.
  3. There are quite a few spelling and grammatical errors which should be proofread and corrected throughout your manuscript.
  4. Overall, it is an interesting manuscript, but research significance, data analyses and discussion are not acknowledged.

Reviewer 3 Report

  • Please explain lines 96-97, page 3 since the meaning can be not clear.
  • About Methods: in relation to aim1, since the two groups number is different and this can represent a bias for the analysis, can Authors Consider to compare the beach group and the scuba diving group A, then to compare the beach group and the scuba diving group B?

Moreover, Authors should consider a statistical way to consider the statistically significant results about variables/activities (first part of table 2, before POMS results).

  • Please note that in table 2, page 5, there is a word highlighted in yellow. Moreover, the name of the last author should be corrected.
  • Could Authors explain the sentence “POMS before the activity for total scores and all categories, regular medication intake and children under 18 (data not shown)”, page 8, line 212?
  • About Discussion: How Authors consider the peculiar environment they describe in relation to the psychological impact of the activities analyzed?

Moreover, the Depression scores can be examined in depth.

Reviewer 4 Report

General comments
I really enjoyed this piece of research and having followed the existing research in relation to scuba diving I believe that this is an area that requires further investigation. I believe that the current research provides a further insight of the potential benefits of scuba diving, however a number of changes in the introduction, discussion and conclusion would enhance the quality of your manuscript.

Specific comments

Abstract: I believe that scuba diving at least for beginning cannot occur without instructors – please clarify

Page 1 line 17: Please provide the p values etc. for the significant differences for those taking medication as this is a very interesting findings that may have multiple practical implication for the future

Introduction:

Page 1 lines 34-36 same as in the abstract please re-structure

Page 2 lines 48-50 if those studies were qualitative then the sample would be quite strong – please provide further details about this – if not then more qualitative studies are necessary

In general more information needed about Deptherapy UK which is amongst the first if not the first initiative examining scuba diving as a potential therapeutic way

Page 2 lines 57-68 instead of the negative aspects I believe it would be better to focus on the potential benefits which will then lead on to the need for further research in this area. The negative consequences could be discussed earlier on in the introduction, however they need to be introduced in a better format as at their current state they are quite descriptive.

Material and methods

Further rationale and explanation is needed for the division of the sample

Line 129 format issue

Results

Information about the sample is required in the methods section rather than the results

Table 2 format issues

I am concerned about the density of the information provided in tables 2 and 3. I would rather see the key information from your data presented as at the moment it is quite difficult for the reader to unravel those tables and understand your data.

The significant differences in POMS and particularly in vigour for those taking medication is in my opinion the key finding of your research

Discussion

Lines 280-283 instead of assuming that a qualitative study may be biased I believe it would be better if you could use it to reinforce your findings as both your current study and Morgan’s study support that scuba diving can have a positive effect on mental health and wellbeing. Since this was your key finding I believe you could focus even more on this aspect in the discussion section which will enhance its quality.

Not knowing if people actually had chronic diseases is a limitation that needs to be clearly described and in addition this is linked with the recommendations for future research as a sample for future research could be only people with chronic diseases for example.

Conclusion

In my opinion the conclusion needs to be re-written in order to highlight the key findings, provide specific suggestions for future research and identify some potential practical implications

Round 2

Reviewer 2 Report

Thank you for your efforts to address my previous comments. I will accept the present version.